# A Study on the Corrosion Behavior of RGO/Cu/Fe-Based Amorphous Composite Coatings in High-Temperature Seawater

**Zhenhua Chu** , **Yunzheng Zhang, Wan Tang, Yuchen Xu and Jingxiang Xu** *

Department of Mechanical Engineering, College of Engineering, Shanghai Ocean University,
Shanghai 201306, China; zhchu@shou.edu.cn (Z.C.); yz_zhang7@163.com (Y.Z.); wtang@shou.edu.cn (W.T.);
17749450088@163.com (Y.X.)
* Correspondence: jxxu@shou.edu.cn; Tel.: +86-22-60204810; Fax: +86-22-26564810

**Abstract:** In this paper, based on an Fe-based amorphous alloy, four kinds of RGO/Cu/Fe-based amorphous composite coatings with mass ratios of 5%, 10%, 15%, and 20% of RGO/Cu were prepared on the surface of 45# steel by using high-velocity oxy-fuel (HVOF) spraying. The coatings were immersed in simulated seawater at room temperature and at 90 °C for different lengths of time, and their corrosion resistance was tested using electrochemical impedance spectroscopy (EIS), scanning electron microscopy (SEM), and X-ray diffraction (XRD), and the surface morphology and phase distribution of the samples were observed. The results showed that with the increase in the introduction ratio of RGO/Cu, when the addition ratio reached 15%, the composite coating had the best corrosion resistance. After soaking in simulated seawater at 90 °C for 18 days, the surface of the coating showed slight peeling and crack propagation, but no obvious pitting phenomenon occurred. The corrosion mechanism of the RGO/Cu/Fe coating in high-temperature seawater is mainly that high temperature causes the cracking of the coating, which opens up a transport channel for corrosion media. However, due to the addition of RGO, the corrosion has a certain self-limitation effect, which is mainly due to the toughening effect of RGO on the coating and its effect on extending the corrosion channel.

**Keywords:** Fe-based amorphous composite coating; graphene; high-temperature seawater; corrosion





## 1. Introduction

As land resources are increasingly depleted, people have turned their attention to the development of marine resources. Human development of the ocean relies on various marine equipment. However, in the process of service, marine equipment is subject to severe corrosion due to the presence of moist, salt-laden air and water vapor in seawater. Marine engineering equipment such as ships and offshore platforms work under severe corrosive environments such as salt spray corrosion (caused by solid NaCl and water vapor) in high-temperature environments, which poses challenges to marine equipment materials [1–3]. Although traditional metal materials have been studied to improve the corrosion resistance, their protective effect is limited in such harsh marine environments [4–6]. In order to improve the durability of such equipment, spraying high-performance coatings has become a common and effective protective method, aiming to form a strong barrier to isolate the direct erosion of corrosive media and metal substrates [7–10].

Compared with metal alloys, amorphous alloys exhibit remarkable advantages in terms of the physical, chemical, and mechanical properties, such as the high strength and hardness, high elastic strain limit, and excellent wear resistance and corrosion resistance. In 2013, Ye and Shin [11] synthesized Fe–Cr–Mo–W–Mn–C–Si–B metallic glass composite materials containing a large amount of amorphous phase using the laser direct deposition method. They found that the microhardness ($HV_{0.2}$ 1591) of the amorphous phase was significantly higher than that of the crystalline phase ($HV_{0.2}$ 947), and the wear resistance

increased significantly with the increase in the amorphous phase ratio. However, due to the limitation of the glass-forming ability (GFA), it is very difficult to fabricate large-scale BMG workpieces or directly use them as structural materials, which limits the application of amorphous alloys. Fortunately, amorphous coatings prepared by thermal spraying have demonstrated their advantages in corrosion resistance [12], showing great potential in industrial applications. Lin et al. [13] prepared $Fe_{40}Cr_{19}Mo_{18}C_{15}B_8$ Fe-based amorphous coating on 316 stainless steel using the high-velocity oxy-fuel (HVOF) spraying method. The experimental results showed that the Fe-based amorphous alloy coating exhibited a transition behavior of activation, passivation, and overpassivation in seawater and 3.5% NaCl solution. After alternating cold and hot salt-spray corrosion and high-speed water erosion tests, although the coating surface showed rusting after 12 weeks, its weight loss was minimal and a stable passive film was formed. This indicates that Fe-based amorphous alloys have excellent stable passivation and corrosion resistance properties. Wang et al. [14] prepared Fe-based amorphous coatings on an AISI 1020 steel tube surface using high-speed laser cladding technology, showing excellent corrosion resistance. The corrosion potential of this coating was as high as $-0.471$ V, the corrosion current density was as low as $2.7 \times 10^{-6}$ A/cm$^2$, and the polarization resistance value was as high as 22,149 $\Omega \cdot$cm$^2$. These excellent corrosion resistance properties are mainly due to the high content of amorphous phase (up to 95%) in the coating and the protection of the Cr oxide layer formed on the surface.

Adding a second phase to Fe-based amorphous alloys is often an effective means of improving their corrosion resistance. Chu et al. [15,16] prepared TiN/Fe-based amorphous composite coatings and AT13/Fe-based amorphous composite coatings through plasma-spraying technology. From the morphology point of view, the typical layered structure of the thermal spray coatings and the close combination of the two phases in these two composite coatings have no obvious structural defects, and they have good corrosion resistance and wear resistance.

In recent years, graphene, as a new emerging material, has attracted worldwide attention for its unique properties and wide application prospects. In the field of corrosion protection, graphene is a cutting-edge material that can be used as a nano-filler reinforcement material to enhance its anti-corrosion performance [17,18]. Various research results reveal that the effectiveness of graphene in preventing corrosion is mainly attributed to its unique "maze effect" [19–21]. When corrosive media attempt to penetrate the graphene structure, its intricate layout makes the diffusion path of the corrosive media extremely tortuous and difficult. In addition, graphene can significantly fill the tiny pores in composite coatings, thereby reducing the porosity of the coating and enhancing its compactness [22]. Moreover, graphene is also known for its excellent mechanical properties. Introducing graphene as an additive into composite coatings can greatly improve the wear resistance and other mechanical properties of the coating [23,24].

In fact, the corrosion damage of high-temperature seawater to materials is more severe. For marine equipment that must operate in high-temperature environments, they need to be exposed to high-temperature seawater for long periods, such as tens or even hundreds of degrees Celsius. For example, the service environment of the armor layer of submarine oil and gas pipelines is generally between 20 °C and 130 °C [25]. Such extreme conditions can easily lead to equipment failure due to corrosion, so there are extremely stringent requirements for its corrosion resistance.

There have been a lot of studies on the corrosion resistance of Fe-based amorphous coatings in ambient seawater [13–16], but there are few research articles on the high-temperature seawater corrosion resistance of Fe-based amorphous coatings. For the high-temperature corrosion resistance of Fe-based amorphous coatings, some scholars believe that high temperatures affect the formation of passive films [26], while other scholars' research has proved the negative impact of high temperatures on metal passive films [27,28]. The working conditions of marine equipment are complex, and a large number of corrosion studies focused on ambient seawater cannot meet the service needs of certain special

equipment, so it is very important to study the corrosion behavior of Fe-based amorphous coatings in high-temperature seawater.

In this study, Fe-based amorphous composite coatings of reduced graphene oxide (RGO)/copper (Cu) were prepared using plasma-spraying technology. The Fe-based amorphous composite coating containing 15% RGO/Cu was immersed in simulated seawater at 90 degrees Celsius for up to 18 days. Its resistance to high-temperature seawater corrosion was comprehensively evaluated and the protective mechanism of the coating was reviewed.

## 2. Experimental Materials and Methods

### 2.1. Preparation of the Coating

GO/Cu composite powder was prepared by the gas-atomizing drying method, in which the mass ratio of GO to Cu was 1:9. Then, the GO in the GO/Cu composite powder was reduced by the thermal reduction method to obtain RGO/Cu composite powder. The Cu powder and Fe-based amorphous powder required for this experiment were purchased from Shanghai Naio Nanotechnology Co., Ltd. (Shanghai, China). The purity of the Cu powder was 99.9%, and the particle size was 1 μm. The chemical formula of the Fe-based amorphous composite powder was $Fe_{45}Cr_{16}Mo_{16}C_{18}B_5$, with a particle size ranging from 15 μm to 45 μm. The GO was purchased from Suzhou Carbon Feng Technology Co., Ltd. (Suzhou, China), with a layer count of 1–2 and a purity of over 98%. The sheet diameter ranged from 0.2 μm to 10 μm. The mechanical mixing of the $Fe_{45}Cr_{16}Mo_{16}C_{18}B_5$ amorphous powder with m (RGO/Cu mass accounting for 5%, 10%, 15%, and 20% of the total) resulted in RGO/Cu/Fe-based amorphous composite powder. Finally, an RGO/Cu/Fe-based amorphous composite coating was prepared on the surface of 45# steel (0.45 wt.% C) with a size of 10 mm × 10 mm × 12 mm by plasma spraying. The powder used for spraying is shown in Figure 1, and the plasma spraying parameters are shown in Table 1 [29].

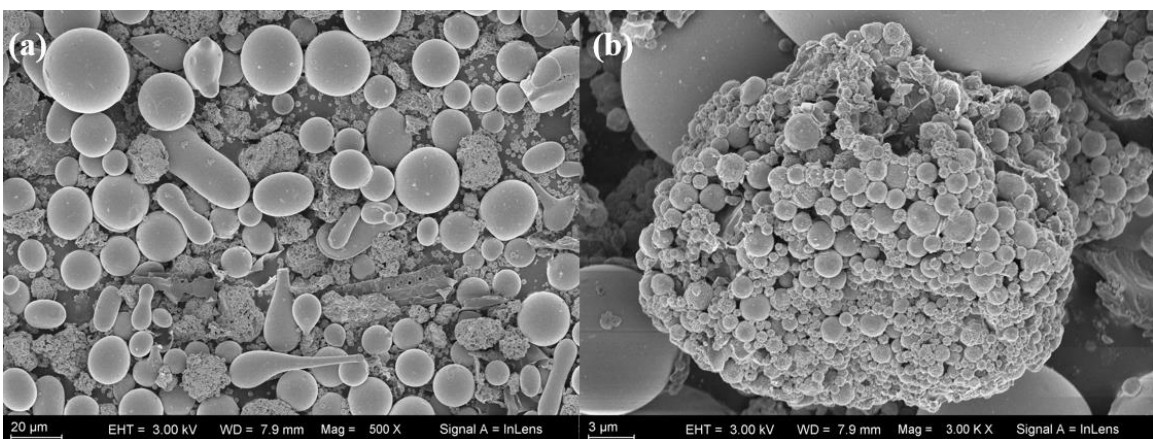

**Figure 1.** RGO/Cu/Fe-based amorphous composite powder: mixed powder (**a**); and single RGO/Cu particle (**b**).

**Table 1.** Spraying process parameters.

| Process Parameters | RGO/Cu/Fe-Based Amorphous Composite Coating |
|---|---|
| Arc voltage (V) | 70 |
| Arc current (A) | 500 |
| Gun distance (mm) | 100 |
| Movement speed of spray gun (m/min) | 5–7 |
| Argon flow rate ($dm^3$/min) | 30 |
| Nitrogen flow rate ($dm^3$/min) | 120 |
| Coating thickness (μm) | 300 |

In order to confirm the introduction of graphene, Raman spectroscopy (as shown in Figure 2a), a conventional method for characterizing graphene, and infrared spectroscopy (as shown in Figure 2b) were used for the composite powder. It can be calculated from the Raman spectrum that the ID/IG value of RGO/Cu was 1.14, and the ID/IG value of GO/Cu was 1.10. After the thermal reduction of the powder, the ID/IG value increased slightly, indicating a slight increase in disorder. The content of various oxygen-containing functional groups is shown in Figure 2b. It can be seen that at 3440 cm$^{-1}$, 1630 cm$^{-1}$, 1400 cm$^{-1}$, and 1060 cm$^{-1}$, oxygen-containing functional groups corresponding to -OH, C=O, C–OH, and C–O–C appeared, respectively. Moreover, the intensity of the oxygen-containing functional groups of the reduced RGO/Cu composite powder decreased, indicating the presence of GO, and GO forms RGO through reduction.

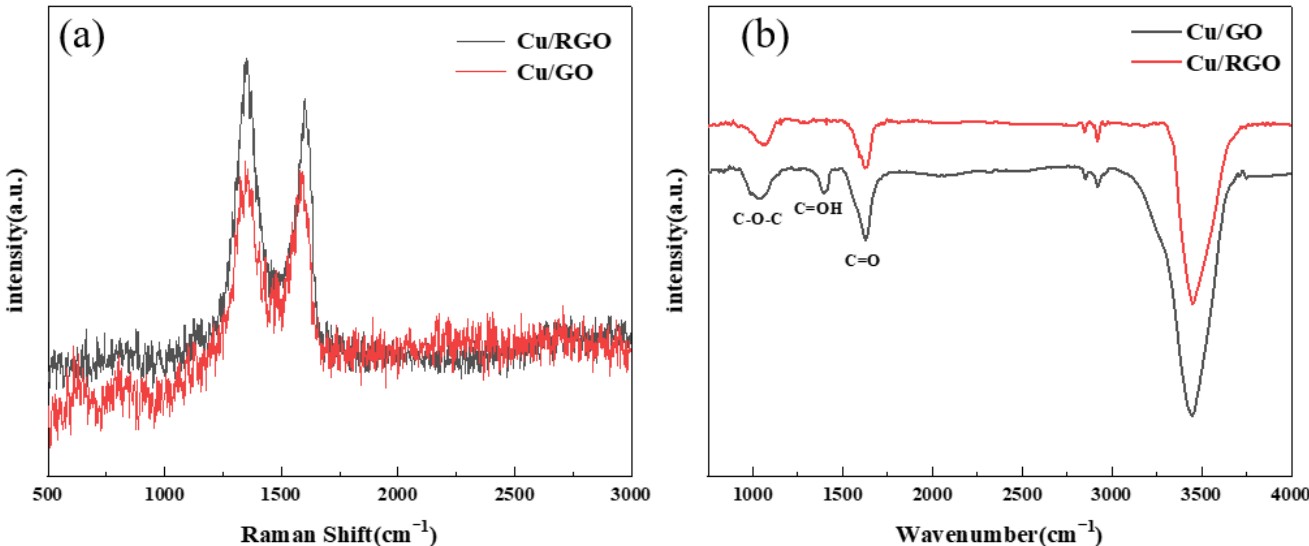

**Figure 2.** Raman spectrum of the composite powder (**a**); and infrared spectrum (**b**).

### 2.2. Experimental Route

In this paper, firstly, potentiodynamic polarization curve scanning was conducted on the coatings with different RGO/Cu addition ratios after being immersed in simulated seawater at room temperature for 30 days. The long-term corrosion performance of the four coatings in simulated seawater at room temperature was compared, and the optimal RGO/Cu addition ratio was obtained. Based on this optimal ratio coating, it was immersed in simulated seawater at 90 degrees Celsius for 18 days, and its high-temperature corrosion performance was tested. A series of characterizations were performed on its microstructure, phase, etc. The simulated seawater composition is shown in Table 2 [15].

**Table 2.** Artificial seawater formula (g/L).

| NaCl | MgCl$_2$ | Na$_2$SO$_4$ | CaCl$_2$ | KCl | SrCl$_2$ | NaHCO$_3$ | KBr | H$_3$BO$_3$ | NaF |
|---|---|---|---|---|---|---|---|---|---|
| 24.530 | 5.200 | 4.090 | 1.160 | 0.695 | 0.025 | 0.201 | 0.101 | 0.027 | 0.003 |

### 2.3. Coating Performance Testing Method

The electrochemical workstation interface 1010E produced by Gamry was used. During the test, a three-electrode system was used, with the coating sample as the working electrode, the saturated calomel electrode as the reference electrode, and the platinum sheet electrode as the counter electrode. Before performing the potentiodynamic polarization curve scan and electrochemical impedance spectroscopy (EIS) experiments, the coating was polished and buffed until it was in a mirror state, and it was then encapsulated with epoxy resin, leaving only a 1 cm$^2$ surface to be tested. The measurement of the potentiody-

namic polarization curve was performed after the open circuit potential (EOCP) stabilized, with a scan rate of 1 mV/s and a scan range set at EOCP ± 500 mV. The test frequency of the electrochemical impedance spectroscopy ranged from 100,000 Hz to 0.1 Hz, with an additional sine wave AC perturbation frequency of 5 mV. The coating was subjected to polarization curve and EIS tests after being immersed in high-temperature simulated seawater for 0 day, 1 day, 4 days, 7 days, and 18 days. Each set of samples was tested three times and averaged to avoid accidental errors. After the testing was completed, data analysis was performed using Gamry Echem Analyst software (version 7.0.0.7). For each sample, the potentiodynamic polarization curve test was conducted after the EIS test.

For the composite coatings with different soaking times, scanning electron microscopy (SEM, S4800, Hitachi, Tokyo, Japan) was used to observe the surface morphology of the coatings. For the composite coatings soaked for 18 days, energy dispersive spectrometer (EDS) was used to analyze the elemental distribution of the coating surface, and X-ray diffraction (XRD, Bruker D8 Focus, Billerica, MA, USA) and X-ray photoelectron spectroscopy (XPS, Thermo Scientific K-Alpha, Thermo Fisher Scientific, Waltham, MA, USA) were used to analyze the phase composition of the coating surface. In the XRD test, the specific scanning angle was 20°–80°, and the scanning speed was 2°/min. In the XPS test, the fine spectra of the Cu and Fe elements were tested.

## 3. Results

### 3.1. Characterization of RGO/Cu/Fe-Based Amorphous Composite Coating

The surface morphology of the composite coatings with different RGO/Cu addition ratios was observed, as shown in Figure 3. As the RGO/Cu ratio increases, the dark-colored portion of the coating surface increases and is more evenly distributed. At the same time, it was observed that all four coating surfaces have a certain porosity, which increases with the increase in the RGO introduction ratio. This may be due to the fact that during plasma spraying, due to the high melting point of RGO, some RGO is not completely melted but is deposited on the substrate surface in a solid form, resulting in an increase in surface defects.

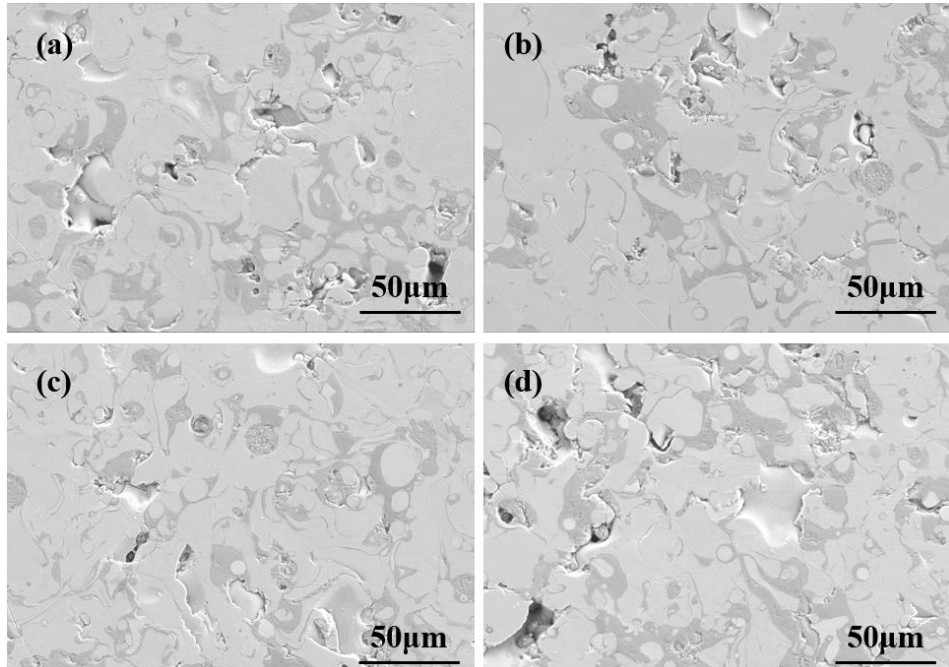

**Figure 3.** Surface morphology of composite coatings with different RGO/Cu addition ratios: 5% RGO/Cu/Fe-based coating (**a**); 10% RGO/Cu/Fe-based coating (**b**); 15% RGO/Cu/Fe-based coating (**c**); and 20% RGO/Cu/Fe-based coating (**d**).

Figure 4 shows the XRD scan results of the RGO/Cu/Fe-based amorphous composite coating after thermal spraying and polishing. It can be seen that the coating forms a typical amorphous diffraction peak, with a relatively high degree of amorphization. The signal peak of Cu is relatively strong. Figure 5 shows the EDS scanning results of the composite coating. The main elements are evenly distributed, and the Cu element has not been agglomerated seriously in a few areas.

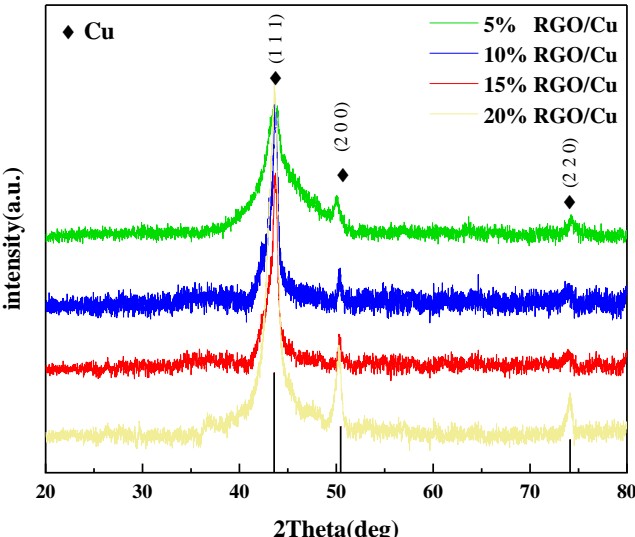

**Figure 4.** XRD scanning results of RGO/Cu/Fe-based amorphous composite coating.

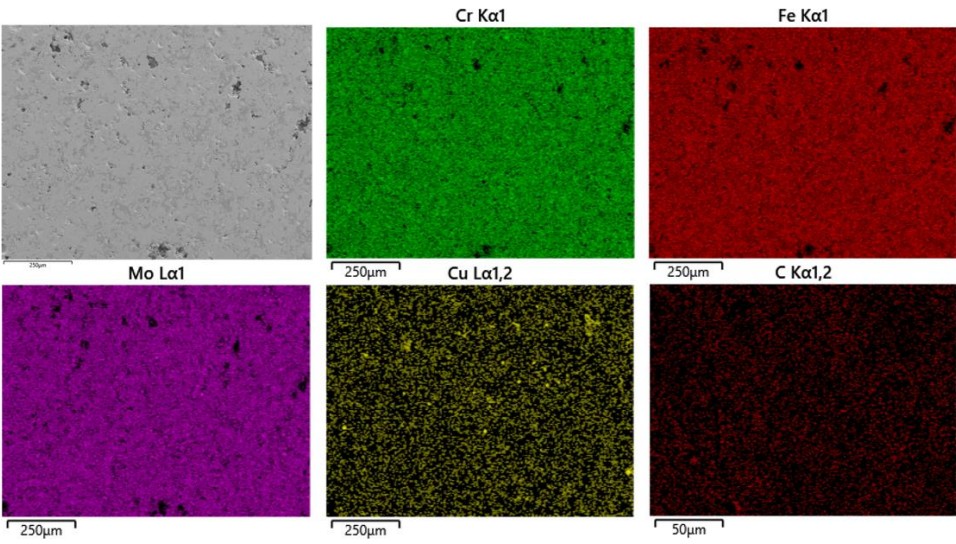

**Figure 5.** EDS results of composite coating.

### 3.2. Analysis of Potentiodynamic Polarization Curve

In this experiment, the content of the powder accounted for 5%, 10%, 15%, and 20% of the total mass of the powder and was named G1/Cu1, G2/Cu2, G3/Cu3, and G4/Cu4, respectively. The potentiodynamic polarization curves of the four composite coatings without immersion are shown in Figure 6a, and the fitting results (as shown in Table 3) indicate that the G3/Cu3 group has the highest self-corrosion potential (−371.3 mV) and the lowest self-corrosion current density (2.22 μA·cm$^2$). After soaking these four composite coatings in simulated seawater at room temperature for 30 days, their potentiodynamic polarization curves were measured (as shown in Figure 6b). The fitting results of the potentiodynamic polarization curves of the four coatings are shown in Table 4. It can be seen that after 30 days of immersion, the G3/Cu3 composite coating has the highest corrosion

potential (−557.8 mV) and the smallest self-corrosion current density (12.04 μA·cm²). It is also found that as the content of RGO/Cu composite powder increases, the corrosion resistance of the coating shows a trend of increasing firstly and then decreasing. Compared with the G3/Cu3 composite coating, the corrosion resistance of the G4/Cu4 composite coating decreases. Considering the influence of porosity on the performance of the coating, as well as the high melting point of RGO, this may be due to the high content of RGO addition causing the thermal spray powder to not melt well, resulting in an increase in porosity in the coating, which in turn affects its corrosion resistance.

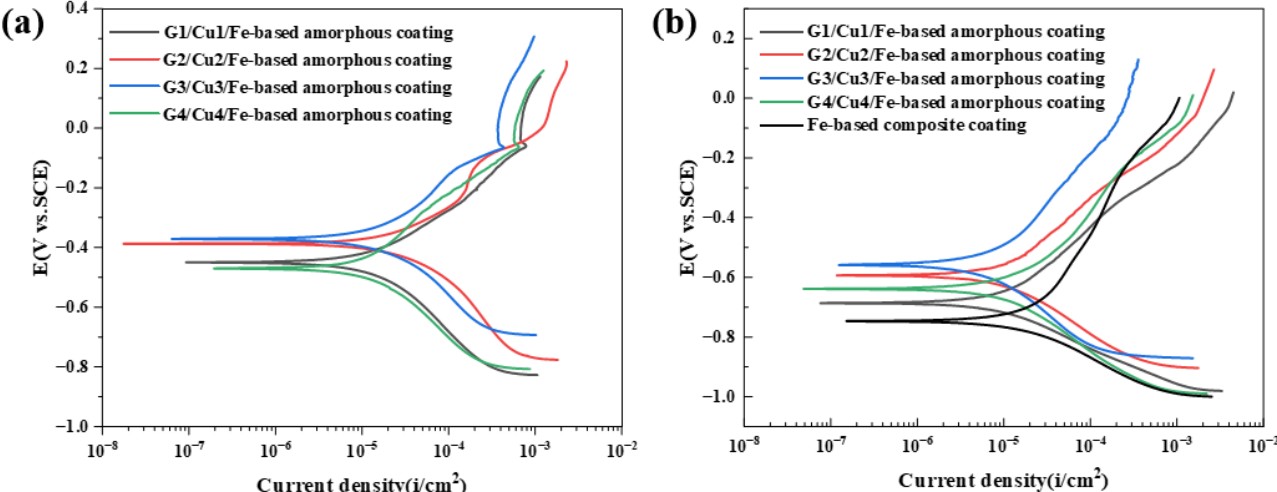

**Figure 6.** Polarization curves of composite coatings with different proportions of added phases after soaking in simulated seawater for 0 day (**a**) and 30 days (**b**).

**Table 3.** Fitting results of dynamic potential polarization curves for composite coatings with different components (soaking in artificial seawater for 0 day).

| Samples | $E_{corr}$ (mV) | $i_{corr}$ (μA·cm⁻²) | CorrRate (mpy) |
|---|---|---|---|
| G1/Cu1/Fe-based amorphous coating | −449.9 | 5.22 | 1.95 |
| G2/Cu2/Fe-based amorphous coating | −387.5 | 2.23 | 0.83 |
| G3/Cu3/Fe-based amorphous coating | −371.3 | 2.22 | 0.83 |
| G4/Cu4/Fe-based amorphous coating | −469.9 | 8.36 | 3.13 |

**Table 4.** Fitting results of dynamic potential polarization curves for composite coatings with different components (soaking in artificial seawater for 30 days).

| Samples | $E_{corr}$ (mV) | $i_{corr}$ (μA·cm⁻²) | CorrRate (mpy) |
|---|---|---|---|
| G1/Cu1/Fe-based amorphous coating | −686.7 | 18.95 | 8.66 |
| G2/Cu2/Fe-based amorphous coating | −593.8 | 16.00 | 7.30 |
| G3/Cu3/Fe-based amorphous coating | −557.8 | 12.04 | 5.50 |
| G4/Cu4/Fe-based amorphous coating | −638.2 | 15.33 | 7.00 |
| Fe-based amorphous coating | −746.5 | 25.58 | 11.69 |

Given that the 15% RGO/Cu/Fe-based amorphous composite coating shows the best corrosion resistance among the four coatings, a high-temperature simulated seawater corrosion experiment was conducted using this coating. The potentiodynamic polarization curves of the 15% RGO/Cu/Fe-based amorphous composite coating immersed in simulated seawater at 90 °C for different days are shown in Figure 7. Meanwhile, the Tafel fitting data of the polarization curves are given in Table 5. It is shown that during the first day of corrosion, the self-corrosion potential of the coating slightly increases, while the self-corrosion current density decreases. At the same time, there is also a significant passivation plateau on both polarization curves, indicating that the coating improves its own corrosion resistance at the beginning stage of the corrosion process. Until the 18th day of immersion, the self-corrosion potential of the coating shows a downward trend, while the magnitude of the self-corrosion current also shows an upward trend, indicating that after the initial stage of corrosion, the corrosion resistance of the coating continues to decrease.

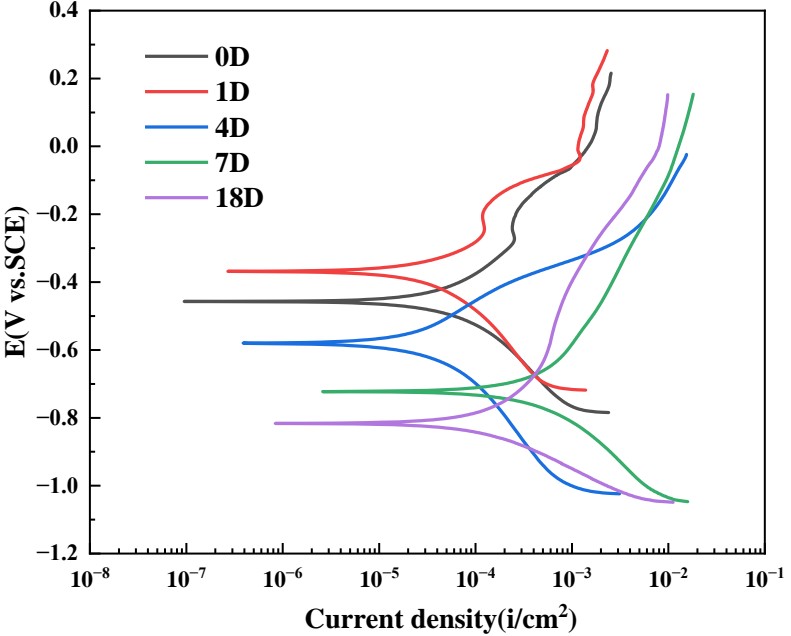

**Figure 7.** Dynamic potential polarization curve of RGO/Cu/Fe-based amorphous composite coating after soaking in simulated seawater at 90 °C for different days.

**Table 5.** Tafel fitting results after soaking in simulated seawater at 90 °C for different days.

| Samples | $E_{corr}$ (mV) | $i_{corr}$ ($\mu A \cdot cm^{-2}$) | CorrRate (mpy) |
|---|---|---|---|
| 0D | −457.2 | 7.65 | 3.50 |
| 1D | −368.2 | 7.41 | 3.39 |
| 4D | −579.8 | 10.19 | 4.66 |
| 7D | −772.6 | 21.64 | 9.89 |
| 18D | −816.7 | 20.12 | 9.19 |

The reason for the increase in corrosion resistance at the beginning of corrosion may be that at the beginning of corrosion, the effect of high temperature on the composite coating is not yet obvious. At the same time, the composite coating undergoes simulated seawater corrosion, and corrosion products are generated at the pores of the coating, forming a passive film on the surface of the coating, blocking the pores, reducing the passage of corrosion media into the interior of the coating through pores and other defects, playing a protective role for the substrate, and reducing the corrosion efficiency of the coating. As the immersion time increases, the coating gradually shows pitting corrosion or even cracks under the influence of high temperature. At this time, the protective effect of the passive

film on the coating begins to weaken until disappearing, resulting in a decrease in the self-corrosion potential and an increase in the self-corrosion current density.

### 3.3. Electrochemical Impedance Spectroscopy Analysis

In order to further investigate the corrosion behavior of the composite coatings in high-temperature seawater, electrochemical impedance spectroscopy was performed on RGO/Cu/Fe-based composite coatings immersed in simulated seawater at 90 °C for different days. Figure 8 shows the Nyquist and Bode plots of the composite coating. From the Nyquist plot, it can be seen that the size of the capacitive arc of the coating after immersion for different days is ranked as follows: 1 D > 0 D > 4 D > 7 D > 18 D. The radius of the capacitive arc increases on day 1, and then the radius of the capacitive arc shows a continuous shrinking trend, indicating that the coating has a passivation phenomenon on the first day, and its corrosion resistance increases in the initial stage of corrosion, but then rapidly decreases, which is also consistent with the trend of changes in the self-corrosion potential and self-corrosion current in the polarization curve.

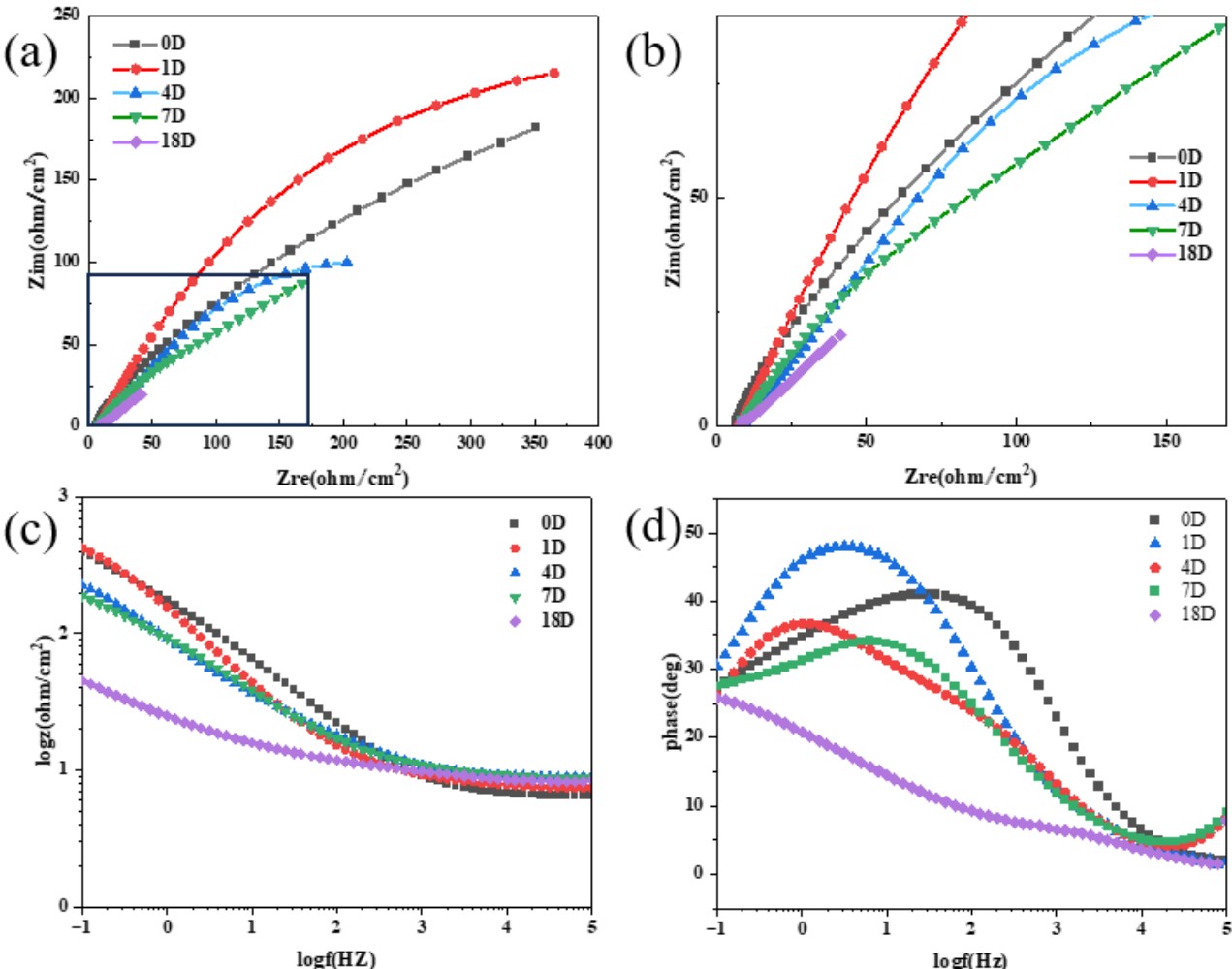

**Figure 8.** 15% RGO/Cu/Fe-based amorphous composite coating soaked in 90 °C simulated seawater for different days: Nyquist plot (**a**,**b**) and Bode diagram (**c**,**d**).

Figure 8c,d show the relationship between the scanning frequency and the impedance mode value and phase angle. As shown in Figure 8d, within the scanning frequency range, there is a peak on the frequency–phase angle diagram for 0 D and 1 D, which represents a time constant. However, for 4 D and 7 D, there is a trend of a second time constant appearing in the high-frequency range. The number of time constants is closely related

to the corrosion behavior. The first time constant represents the nature of the coating, such as passivation, while the emergence of the second time constant often indicates the failure of the coating. Combined with the Nyquist diagram, it can be seen that the coating maintains a passive state during the initial soaking period. After the initial period, the coating begins to exhibit surface defects, which continuously reduce its protective ability. In the frequency–phase angle diagram, it appears that a low-frequency time constant appears on the 18th day for the coating in the mid-frequency range, which may indicate that some significant defects appear on the coating surface at later stages of high-temperature erosion.

The fitting results of the composite coating circuit after soaking in high-temperature seawater for different days are shown in Figure 9, and the fitting data of each component are shown in Table 6. The circuit codes of the three circuits are Rs(CPEf(Rf(CPEdlRct))), Rs(CPEfRf)(CPEdlRct), and Rs(CPEf(Rf(CPEdlRct)))W. It can be seen that Rct shows a trend of first rising and then falling, with its value reaching the highest at 1 D. This may be due to the intense corrosion process, where the corrosion products continue to precipitate, providing a certain degree of protection to the coating, and then start to drop sharply, which may be due to the stress cracking of the coating due to high-temperature heating at this time. The Warburg impedance appearing in the later fitting circuit may also be due to this reason. Until day 7, the Rct value on day 18 is basically the same as that on day 7 due to a balance between corrosion cracking and corrosion products clogging the cracks, at which point the coating still has a certain degree of protection. Rf shows an unstable fluctuation trend, which should be related to the formation and dissolution of surface corrosion products. In this fitting circuit, CPEf represents the coating capacitance, which is related to the coating thickness, surface roughness, and degree of corrosion. Generally speaking, the thicker the coating, the lighter the degree of corrosion and the higher the value. In this fitting result, the value shows a trend of first increasing, then decreasing, and then increasing again. Combined with the fitting results of the potentiodynamic polarization curve and SEM results, this may also reflect the changing trend of the degree of corrosion on the coating surface. CPEdl represents the double-layer capacitance, which is related to the charging and discharging ability of the matrix-coating electric double-layer capacitance and the electrolyte ion concentration. In this fitting result, the value changes in a trend of first increasing and then decreasing, which reflects the process of accumulation–dissolution of corrosion products on the coating surface.

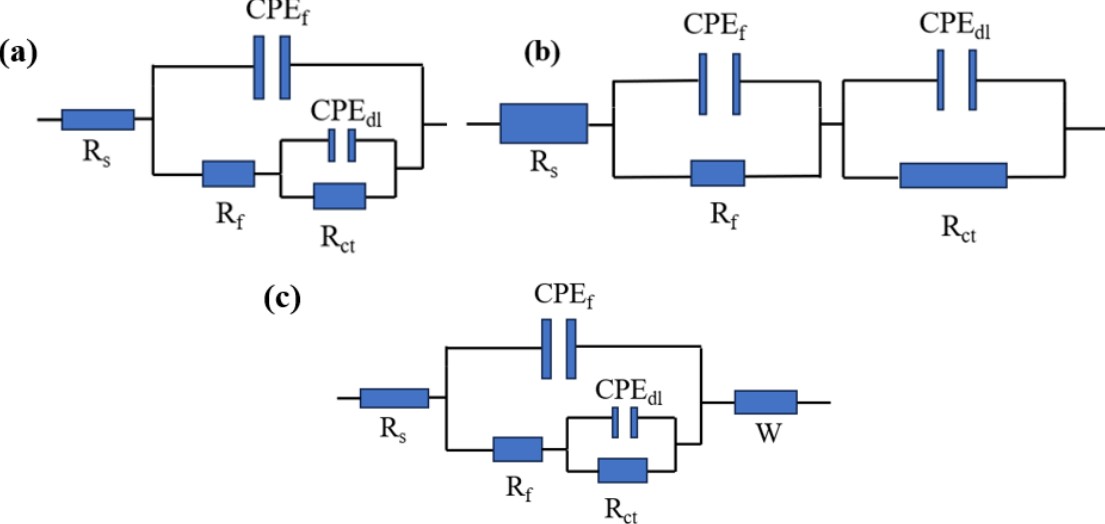

**Figure 9.** RGO/Cu/Fe-based composite coating circuit fitting results after soaking in simulated seawater at 90 °C for different days: 1 D (**a**); 0 D and 4 D (**b**); 7 D and 18 D (**c**).

**Table 6.** Fitting results of various components.

| Immersion Time | $R_s$ ($\Omega \cdot cm^2$) | $CPE_f$ (s-sec$^n$) | $R_f$ ($\Omega \cdot cm^2$) | $CPE_{dl}$ (s-sec$^n$) | $R_{ct}$ ($\Omega \cdot cm^2$) | Chi-Squared |
|---|---|---|---|---|---|---|
| 0D | $6.18 \times 10^0$ | $1.46 \times 10^{-3}$ | $6.41 \times 10^1$ | $2.36 \times 10^{-3}$ | $6.36 \times 10^2$ | $3.35 \times 10^{-4}$ |
| 1D | $7.25 \times 10^0$ | $1.48 \times 10^{-3}$ | $8.48 \times 10^1$ | $3.95 \times 10^{-3}$ | $8.04 \times 10^2$ | $6.10 \times 10^{-5}$ |
| 4D | $8.50 \times 10^0$ | $3.44 \times 10^{-3}$ | $1.81 \times 10^1$ | $3.80 \times 10^{-3}$ | $3.68 \times 10^2$ | $8.18 \times 10^{-4}$ |
| 7D | $1.07 \times 10^0$ | $2.56 \times 10^{-4}$ | $8.53 \times 10^0$ | $4.10 \times 10^{-3}$ | $1.18 \times 10^2$ | $4.19 \times 10^{-4}$ |
| 18D | $7.87 \times 10^0$ | $3.25 \times 10^{-3}$ | $3.50 \times 10^0$ | $2.86 \times 10^{-3}$ | $1.02 \times 10^2$ | $2.87 \times 10^{-4}$ |

### 3.4. Surface Morphology Analysis

To further investigate the performance of the RGO/Cu/Fe-based amorphous coating in high-temperature corrosive environments, the surface of the coating after soaking for different days was carefully observed. The observation results are shown in Figure 10. At the beginning of the soaking period, on the first day, the surface of the coating is smooth, with only a small amount of flaky corrosion products scattered on it. However, as the soaking time extended, from the fourth day onwards, the number of corrosion products shows a significant increase trend. Under high magnification, it can be clearly observed that small cracks have begun to appear on the surface of the coating, which is caused by the coating cracking phenomenon in high-temperature environments, indicating that the protective ability of the coating has begun to decrease. By the seventh day, the increase in corrosion products and the expansion of cracks are more evident, and the protective performance of the coating is further weakened. When the soaking time reached the 18th day, the corrosion condition on the surface of the coating did not change significantly, except for a slight increase in the number of corrosion products and cracks. This series of observation results indicate that in high-temperature corrosive environments, the protective ability of the RGO/Cu/Fe-based amorphous coating gradually decreases over time, but after prolonged soaking, the corrosion and self-healing abilities of the coating can reach a balance, preventing further corrosion of the coating.

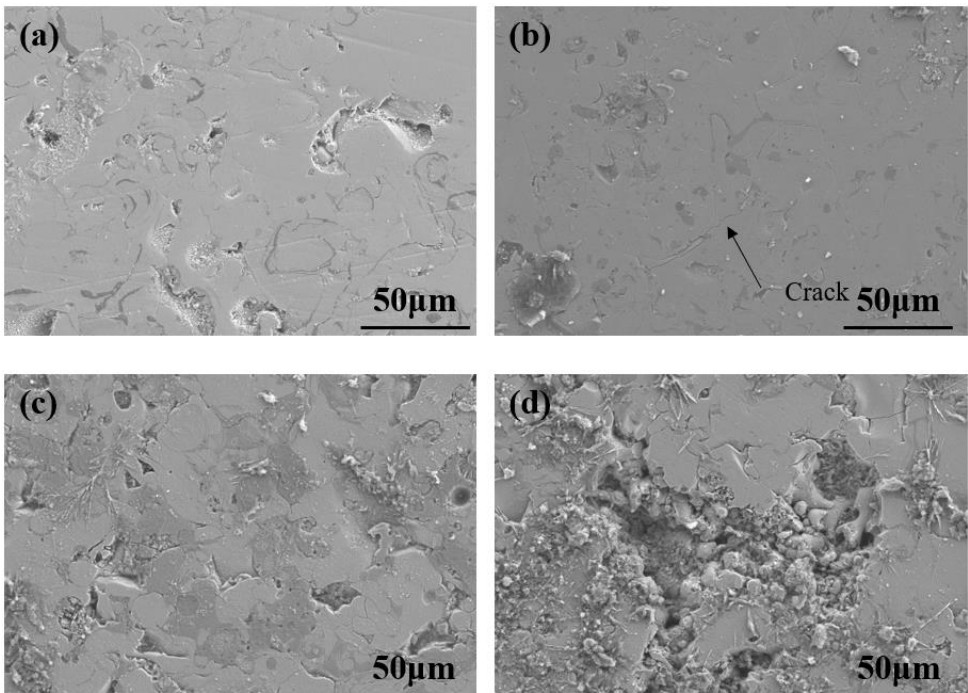

**Figure 10.** Surface morphology of RGO/Cu/Fe-based amorphous coatings soaked in simulated seawater for different days: 1 D (**a**), 4 D (**b**), 7 D (**c**), and 18 D (**d**).

In order to observe the damage degree of the coating surface more intuitively, the RGO/Cu/Fe-based composite coating immersed in high-temperature simulated seawater for different days was observed, as shown in Figure 11. Compared to the pitting failure mode in normal-temperature seawater, the failure mode of the coating in high-temperature seawater is mainly cracking and surface peeling. Except for the good surface morphology of the coating on the first day, from the fourth day onwards, small cracks and small areas of peeling begin to appear on the coating surface. At this point, the protective effect of the coating begins to decrease gradually. The appearance of cracks creates conditions for corrosion agents to penetrate the interior of the coating, forming micro-battery structures locally and accelerating further corrosion of the coating. As the corrosion process progresses, from the seventh day onwards, the cracks gradually increase and expand, while the degree of peeling increases. At this point, the protective effect of the coating is relatively low, and until the eighteenth day, the degree of crack expansion and surface peeling does not significantly deteriorate, indicating that the failure process of the coating is close to completion. This is consistent with the results of the electrochemical experiments, which show that the coatings will undergo a passive state during the initial stage of high-temperature corrosion, then lose their protective effect and finally tend toward a stable corrosion stage. Meanwhile, Figure 12 shows the surface roughness of the coating before and after high-temperature seawater corrosion. As shown in Figure 12b,d, it can be seen that after 18 days of high-temperature seawater corrosion, the surface roughness of the coating increases, reflecting the damage to the coating surface caused by high-temperature corrosion and the accumulation of corrosion products.

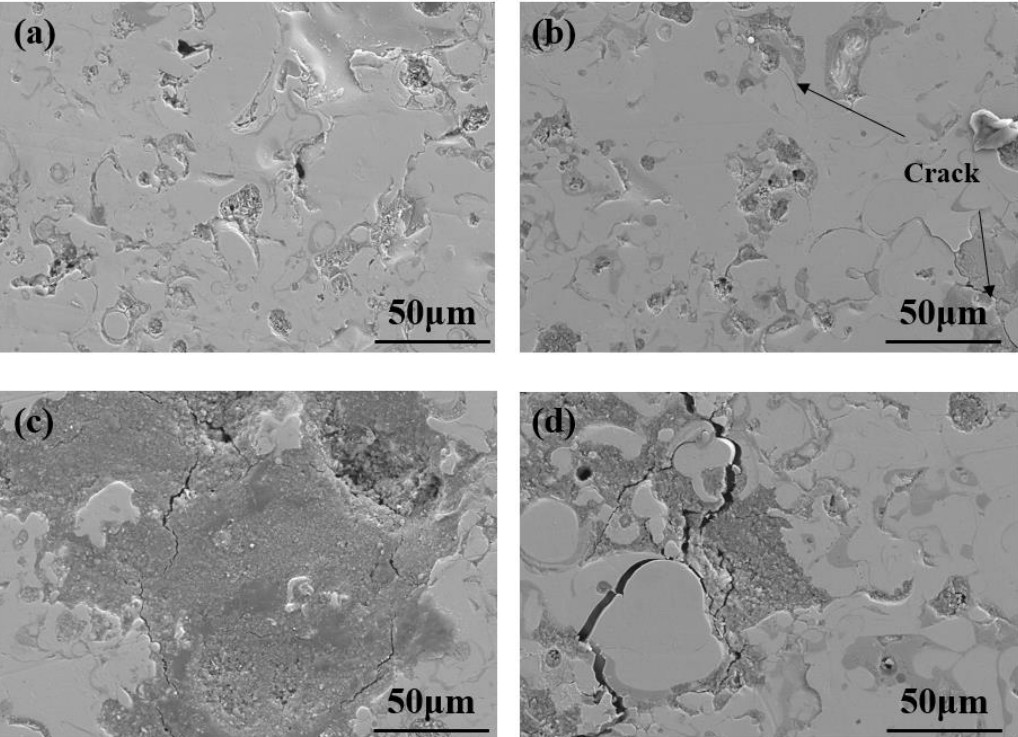

**Figure 11.** Surface morphology of RGO/Cu/Fe-based amorphous coatings after soaking in simulated seawater for different days and removing corrosion products: 1 D (**a**), 4 D (**b**), 7 D (**c**), and 18 D (**d**).

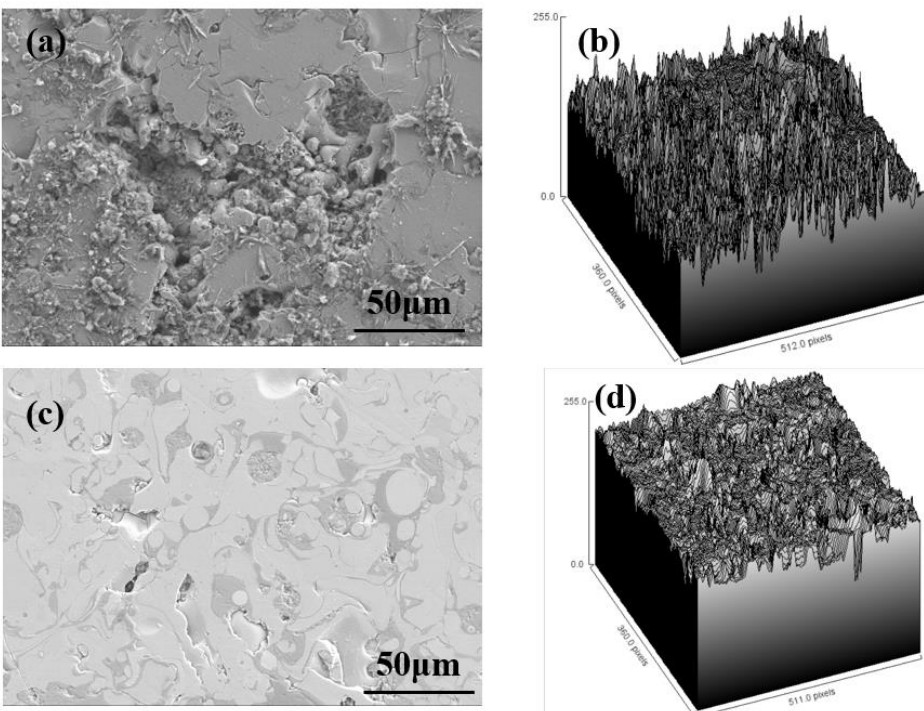

**Figure 12.** Schematic diagram of the surface roughness of the coating after different days of corrosion: surface morphology (**a**) and roughness (**b**) after 18 days of corrosion; and surface morphology (**c**) and roughness (**d**) after 0 days of corrosion.

To determine the distribution of elements on the surface of the coating after high-temperature corrosion, an EDS scan was performed on its surface, as shown in Figure 13. Through observation, it is seen that the coating surface has peeled off after 18 days, and as the surface coating peels off, the metal elements also escape, which causes microscopic galvanic corrosion. The coating defects also occur in the initial pitting or cracks and gradually expand, slowly reducing the protective ability of the coating.

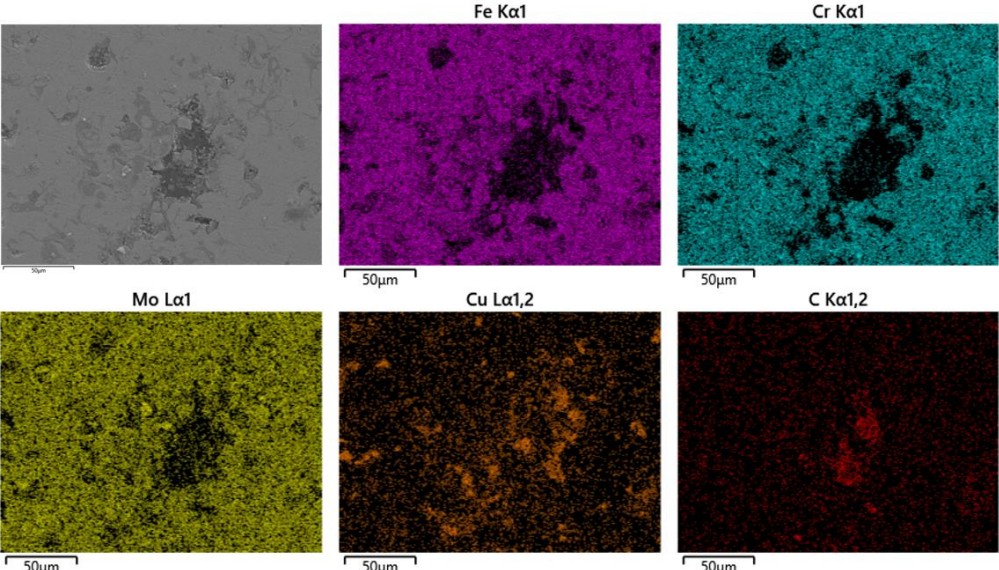

**Figure 13.** EDS image of RGO/Cu/Fe-based amorphous composite coating after immersion in high-temperature seawater for 18 days.

### 3.5. Phase Analysis of Corrosion Products

To further investigate the high-temperature seawater corrosion behavior of the composite coatings, XRD scanning was performed on the surface of the coating immersed in high-temperature simulated seawater for 18 days to determine the corrosion products generated on its surface. The XRD scanning results are shown in Figure 14. It can be seen that after prolonged immersion, the main corrosion product is $Fe_3O_4$, accompanied by a small amount of CuO and FeO(OH). Similar to the corrosion products in normal-temperature seawater, some $Fe_3O_4$ is generated. It is worth noting that the bimetallic interface exhibits special properties in catalytic reactions. Some studies have shown that reactive metal oxides are formed through surface recombination in bimetallic nanoparticles [30–32], demonstrating the interaction between metals, which may include electron transfer, synergistic effects, etc., which may affect the oxidation tendency of the interface, making one of the metals in the bimetallic interface more susceptible to oxidation. Some scholars have studied the role of $SiO_2$-supported AuCu nanocatalysts in CO oxidation reactions [33]. The study showed that under CO and $O_2$ oxidation conditions, the surface of oxygen-pretreated AuCu nanoparticles undergoes atomic recombination, in which $Cu_2O$ forms clusters of CuO. The XRD scanning results for the coating also show the formation of metal oxides such as CuO and $Fe_3O_4$, which may indicate that the formation of corrosion products in the composite coating is related to this bimetallic interface.

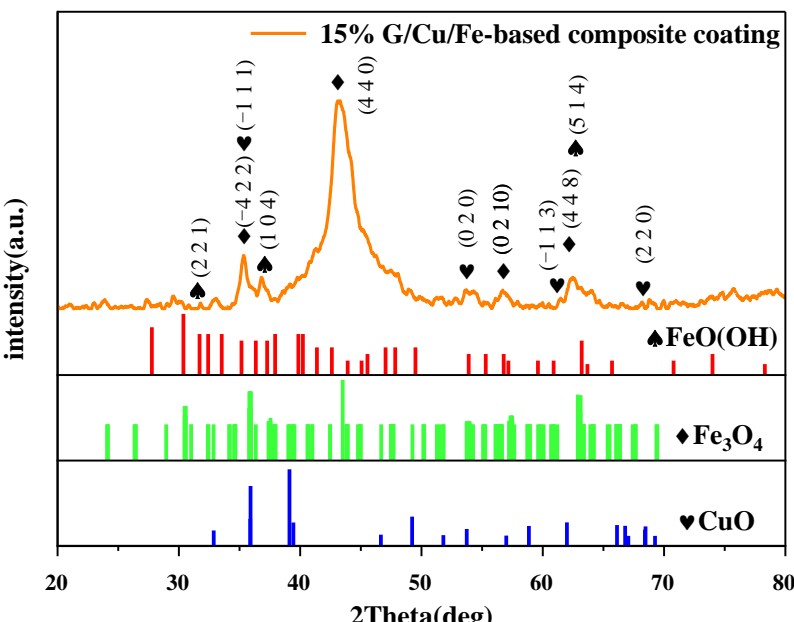

**Figure 14.** XRD scan results of 15% RGO/Cu/Fe-based amorphous coating soaked in high-temperature simulated seawater for 18 days.

To further confirm the formation of corrosion products on the surface of the coating, XPS scanning was performed on the surface of the composite coating after soaking in high-temperature simulated seawater for 18 days. The scanning results concerning the Cu and Fe elements are shown in Figure 15a,b, respectively. In the scanning result for the Fe-2p orbital, there are three valencies of the Fe element: 0 valency, 2 valency, and 3 valency. It can clearly be seen that the peak area of the 2-valent Fe is the largest, followed by the 3-valent Fe, and the peak area of the 0-valent Fe is very small. The curve only shows a small fluctuation near 714 eV, and satellite peaks of 2-valent Fe and 3-valent Fe appear in the curve, confirming the reliability of the scanning results. Combined with the XRD phase results, there are phases of $Fe_3O_4$ and Fe on the surface of the coating, and the valencies in the compound can correspond to this scanning result.

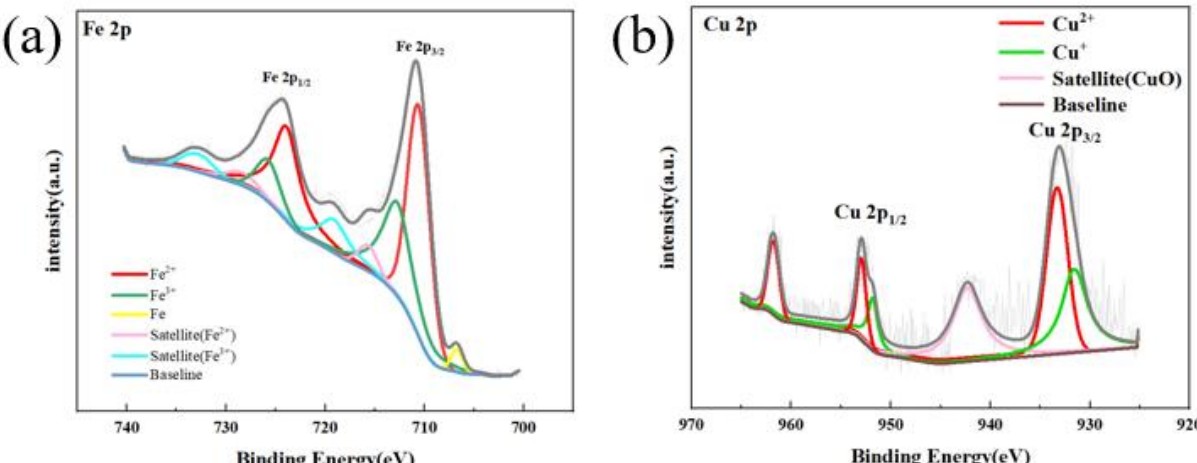

**Figure 15.** XPS scan results of composite coatings soaked in high-temperature simulated seawater for 18 days. Fe-2p (**a**); Cu-2p (**b**).

A large area of $Cu^{2+}$ peaks and a small area of $Cu^{+}$ peaks appears on the Cu-2p orbital. After a period of high-temperature corrosion, the Cu element is oxidized. Combined with the XRD scanning results, there is a certain amount of CuO in the phase, and the XRD peak intensity also corresponds to the peak area in the XPS results. The satellite peaks of CuO are also shown in the XPS diagram, which confirms the reliability of the test results.

## 4. Discussion

In order to further reveal the corrosion mechanism of RGO/Cu/Fe-based amorphous composite coatings, a schematic diagram is shown in Figure 16. During the spraying process, it is inevitable that some pores will appear in the single Fe-based amorphous coating, even if the spraying quality of the coating is excellent. Pitting defects are prone to occur and expand at the pores. In high-temperature seawater, the difference in thermal expansion coefficients between different phases can easily lead to the expansion and cracking of the coating, resulting in the corrosion medium reaching the coating–substrate interface through defects [34]. Copper in the composite coating is beneficial for sealing the pores generated by thermal spraying, reducing the stress concentration at the pores, and reducing the occurrence of pitting corrosion. This may be why there are few pitting corrosion defects on the surface of the coating after removing the corrosion products in the SEM images. Graphene can increase the toughness of the coating and hinder the generation of cracks during exposure to high-temperature environments [35]. At the same time, due to its "maze effect", the corrosion medium is difficult to penetrate through graphene to reach the substrate, which is beneficial for forming a physical barrier, extending the corrosion channel, and prolonging the service life of the coating.

In high-temperature seawater, the following chemical reactions mainly occur on the surface of the coating:

$$4Fe + 3O_2 + 2H_2O = 4FeO(OH) \tag{1}$$

$$2Fe + O_2 + 2H_2O = 2Fe(OH)_2 \tag{2}$$

$$4Fe(OH)_2 + O_2 + 2H_2O = 4Fe(OH)_3 \tag{3}$$

$$2Fe(OH)_3 \rightarrow Fe_2O_3 + 3H_2O \tag{4}$$

$$2Fe_2O_3 + 2Fe + O_2 = 2Fe_3O_4 \tag{5}$$

$$2Cu + O_2 + 2H_2O \rightarrow 2Cu(OH)_2 \tag{6}$$

$$Cu(OH)_2 \rightarrow CuO + H_2O \tag{7}$$

In high-temperature seawater, the high temperature accelerates the dissolution of the corrosion product films $Fe(OH)_2$ and $Cu(OH)_2$, resulting in the formation of more $Fe_3O_4$ and $CuO$ on the coating surface, which is also consistent with the phase analysis results.

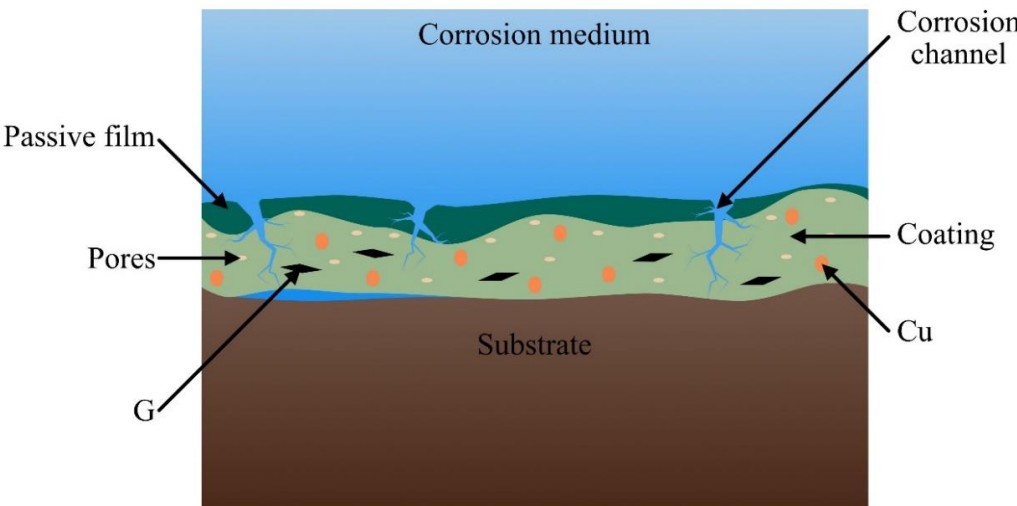

**Figure 16.** Schematic diagram of the corrosion behavior of RGO/Cu/Fe-based amorphous composite coating in high-temperature seawater.

## 5. Conclusions

This paper studies the corrosion behavior of RGO/Cu/Fe-based amorphous composite coatings under simulated seawater at 90 °C, and reaches the following conclusions:

(1) With the increase in the proportion of RGO/Cu introduced, the corrosion resistance of the coating increases firstly and then decreases, with the best result occurring at a doping ratio of 15 wt.%.

(2) The main failure modes of Fe-based amorphous composite coatings under high-temperature seawater are coating cracking and peeling, with limited pitting corrosion.

(3) The introduction of RGO can effectively increase the toughness of the coating, improve its mechanical properties, and help to suppress the propagation of cracks at high temperatures.

(4) In high-temperature seawater, the corrosion products on the coating will block cracks and pores, and the corrosion rate will decrease after reaching a maximum value, as a result of the balance between the corrosion and the blocking effect of the corrosion products.

**Author Contributions:** Formal analysis, Z.C., W.T. and J.X.; Investigation, Z.C.; Writing—original draft, Z.C. and Y.Z.; Writing—review & editing, Z.C. and Y.X. All authors have read and agreed to the published version of the manuscript.

**Funding:** This research was funded by the National Nature Science Foundation of China (No. 51872072) and by special funding for the development of science and technology of Shanghai Ocean University (No. A2-0203-00-100231 and A2-2006-00-200371).

**Institutional Review Board Statement:** Not applicable.

**Informed Consent Statement:** Not applicable.

**Data Availability Statement:** Data are contained within the article.

**Conflicts of Interest:** The authors declare no conflict of interest.

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
