# Peer review of "A Study on the Corrosion Behavior of RGO/Cu/Fe-Based Amorphous Composite Coatings in High-Temperature Seawater"

_coatings, doi:10.3390/coatings14050556_

Round 1

Reviewer 1 Report

Comments and Suggestions for Authors

The research is performed at a high professional level. The authors apply modern analytical instruments, refer to relevant sources. The quality of interpretation of the results and the reliability of the conclusions drawn do not cause doubts. The article can be published with the following comments:

1. Provide data on the parameters and suppliers of the materials and equipment used.

2. The parameters of the analytical methods used in "2.3 Coating performance testing method" should be carefully described.

3. A larger scale bar should be added to the figures presenting the SEM results.

4. The indexes on the XRD scan results should be indicated, as well as the # PDF.

5. Figures 9-10 would be interesting to see additionally at higher magnification. The additional window in Figure 9 (b) is unreadable.

6. In section "4. Discussions" the authors provide a schematic diagram of corrosion behavior of RGO/Cu/Fe-based amorphous composite coating. The paper would have benefited if the authors had provided the same data obtained by SEM on cross sections of the studied coatings.

Author Response

Thank you for all your comments and suggestions. Please see the attachment for detailed responses

Reviewer 2 Report

Comments and Suggestions for Authors

Author Response

(The authors gave the same response as above.)

Reviewer 3 Report

Comments and Suggestions for Authors

This paper reports the experimental results of the corrosion resistance of composite coatings made from Fe-based amorphous alloy, graphene oxide-reduced graphene oxide (RGO), and copper (Cu) on steel surfaces. Different ratios of RGO/Cu/Fe-8 composite coatings were prepared using high-velocity oxy-fuel (HVOF) spraying. These coatings were then subjected to immersion in simulated seawater at room temperature and 90°C for various durations. Corrosion resistance was evaluated using electrochemical impedance spectroscopy (EIS), scanning electron microscopy (SEM), and X-ray diffraction (XRD). The findings revealed that increasing the proportion of RGO/Cu in the composite coatings up to 15% resulted in optimal corrosion resistance. After immersion in 90°C simulated seawater for 18 days, slight peeling and crack propagation occurred on the coating surface, but no significant pitting was observed. The corrosion mechanism in high-temperature seawater involves coating cracking due to heat, creating channels for corrosion media. I think the study was carefully carried out, and the presented results shows the intriguing aspects of corrosion properties of alloy-RGO composite. I think the papers can be improved by considering the following points.

1.     Exploring the origins of bimetallic synergy involving Cu and Fe, which enhances corrosion resistance, is a point of interest. Both Cu and Fe undergo significant oxidation, leading to the formation of interfaces like Cu-FeOx, Fe-CuOx, or FeOx-CuOx. This aspect has been extensively discussed within the catalysis community and investigated in previous works [See, for instance, C. H. Wu et al., Nat Catalysis, 2, 78–85 (2019); T. S. Kim et al., Applied Catalysis B: Environmental 331, 122704 (2023); Y. Pan et al., ACS Appl. Energy Mater. 4, 11151–11161 (2021); Y. Song et al., ACS Catalysis 13, 13777 (2023)]. Incorporating these findings with proper citation into the current investigation would enhance its relevance.

2.     Presenting morphological changes in the samples to illustrate alterations in surface roughness before and after corrosion processes is essential. This can be achieved through scanning electron microscopy (SEM) or atomic force microscopy (AFM) analysis.

3.     While experiments were conducted under seawater conditions, performing the same experiment under pure water conditions for reference could provide valuable insights.

4.     The quality of reduced graphene oxide (RGO), including defect levels, plays a critical role in determining the corrosion resistance of alloy-RGO composites. This aspect can be addressed through Raman spectroscopy analysis, which can also be included in the presentation of results.

Comments on the Quality of English Language

 Minor editing of English language required

Author Response

(The authors gave the same response as above.)

Round 2

Reviewer 2 Report

Comments and Suggestions for Authors

Everything corrected, accept in present form

Reviewer 3 Report

Comments and Suggestions for Authors

The authors' response to my comments and criticism is satisfactory, therefore, I suggest the paper be accepted as it is.